# Effect of Lactic Acid Bacteria Fermentation on Plant-Based Products

**Xiaohua Yang, Jiaqi Hong, Linhao Wang, Changyu Cai, Huanping Mo, Jie Wang, Xiang Fang** [ID] **and Zhenlin Liao ***

College of Food Science, South China Agricultural University, Guangzhou 510642, China;
y-xiaohua@stu.scau.edu.cn (X.Y.); jiaqi@stu.scau.edu.cn (J.H.); 20213164064@stu.scau.edu.cn (L.W.);
caichangyu@stu.scau.edu.cn (C.C.); magicvial@stu.scau.edu.cn (H.M.); jiewang@scau.edu.cn (J.W.);
fxiang@scau.edu.cn (X.F.)
* Correspondence: larryliao@scau.edu.cn; Tel.: +86-186-8023-8136

**Abstract:** Lactic acid bacteria effectively utilize the nutrients and active compounds in plant-based materials via their powerful metabolic pathways and enzyme systems, achieving a combination of nutrition, functionality, and deliciousness. Currently, the majority of review articles predominantly concentrate on summarizing the fermentation of fruits and vegetables by lactic acid bacteria, devoting comparatively less attention to researching other plant species varieties and plant-based by-products. Furthermore, the summary of the research on the active substances and functional properties lacks sufficient depth. This review provides a comprehensive overview of the status of and technological progress in lactic acid bacteria fermentation of various plant species and plant-based by-products, and the effects of lactic acid bacteria on the active substances and functional properties are emphasized. In addition, this review emphasizes that active substances give products more functionality. The aim of this review is to emphasize the significant contribution of lactic acid bacteria to the active substances and functional properties of plant-based products, which will assist researchers in better comprehending the application value of lactic acid bacteria in the plant-based domain and direct attention towards the interaction mechanisms between active substances and product functionality. Concurrently, this review provides a certain theoretical foundation and reference for the application of fermented functional products in promoting health and preventing diseases.

**Keywords:** lactic acid bacteria; plant-based products; fermentation process; active substances; functional properties

## 1. Introduction

Lactic acid bacteria have a well-established history of application in the fermentation of various food products, with approximately 80% of their usage concentrated in the fermentation of dairy products [1]. However, the development of dairy products in recent years has faced certain challenges, such as the high cholesterol and saturated fat content of dairy products, which can increase health risks [2]. Furthermore, the expansion of livestock farming results in a significant production of greenhouse gases. Against this background, more and more people are beginning to pay attention to veganism [3]. As a result, plant-based products have become a new development trend [4]. Plant-based materials comprise significant amounts of carbohydrates, phenolic compounds, vitamins, and other substances. Lactic acid bacteria can hydrolyze large molecules that are difficult to digest into small molecules that are easily absorbed by the human body, or can degrade and transform to produce new organic acids, phenolic substances, volatile substances, and other active substances. The alteration of active substances is a key factor in giving specific functionality to plant-based products. For example, lactic acid bacteria produce acidic metabolites during the fermentation process, which can protect active substances such as vitamins and phenols in plant-based materials, or convert into other polyphenols, such as catechins and anthocyanins, resulting in fermented products with enhanced antioxidant activity [5]. Lactic acid bacteria can metabolize and transform complex, indigestible proteins, cellulose,

and other substances in plant-based materials to produce amino acids, volatile substances, organic acids, and other active ingredients, thereby improving the flavor of fermented products [6]. Lactic acid bacteria can use the rich nutrients in plant-based materials to produce organic acids, bacteriocins, and other active ingredients to enhance the antibacterial ability of products [7]. In addition, after plant-based materials are processed into products, plant-based by-products (such as peel, pulp residue, etc.) cause serious pollution in the environment. However, these plant-based by-products have more nutrients and effective ingredients, but their utilization rate is low due to their difficult processing and poor flavor. The fermentation of lactic acid bacteria can solve the above problems and transform plant-based by-products into valuable products. In conclusion, lactic acid bacteria fermentation is of great significance for plant-based products. The effect of lactic acid bacteria fermentation on plant-based products is described in Figure 1. Lactic acid bacteria fermentation provides a new way to develop plant-based products with high nutrition and functionality.

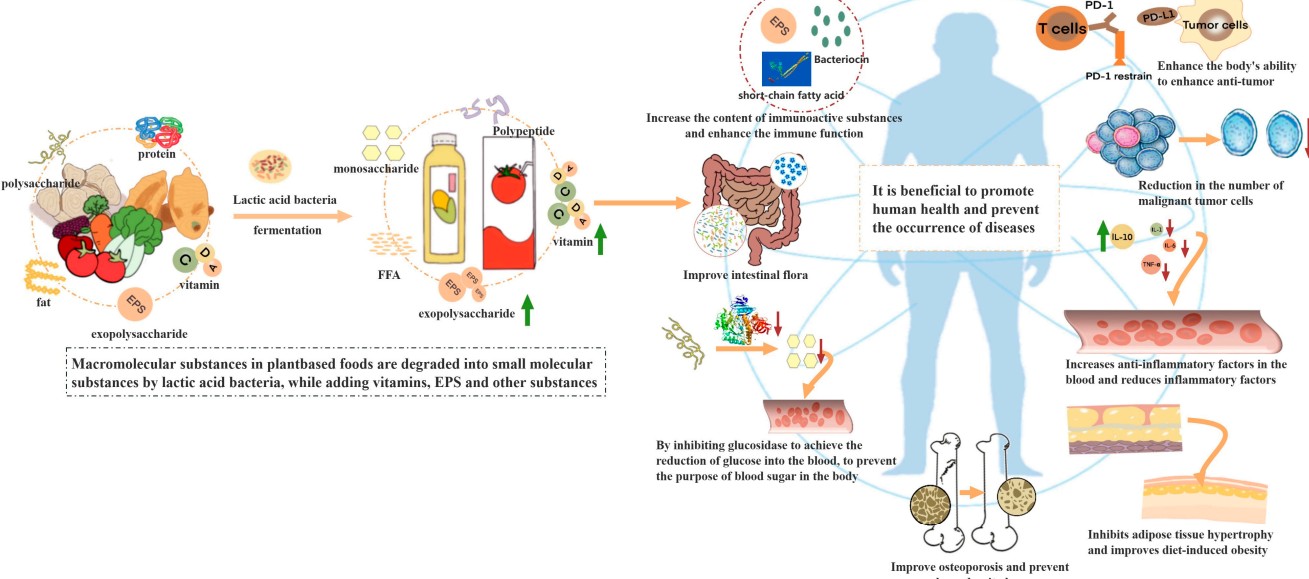

**Figure 1.** The effect of lactic acid bacteria fermentation on plant-based products. (FFA: free fatty acid; T cells: T-lymphocyte; PD-L1: Programmed cell death 1 ligand 1).

This review systematically introduces the applications and technologies of lactic acid bacteria in the field of plant-based products, aiming to enhance researchers' understanding of the potential of lactic acid bacteria fermentation in this area. Secondly, this review focuses on the impact of lactic acid bacteria fermentation on the active ingredients and functional properties of plant-based products. It aims to highlight the significant potential of lactic acid bacteria in the development of functional products, as well as to promote the application of fermented products in disease prevention. Finally, this review identifies the current challenges associated with lactic acid bacteria fermentation of plant-based products. The aim is to promote the development, application, and innovation of fermented plant-based products.

## 2. The State of Lactic Acid Bacteria Fermentation on Plant-Based Products

Lactic acid bacteria are probiotics that can use carbohydrates to produce organic acids, exopolysaccharides, and other substances, which can have beneficial effects on host health. Lactic acid bacteria demonstrate a potent metabolic capacity and enzymatic system, allowing them to decompose proteins, sugars, vitamins, and other substances found in plant-based materials [8]. Consequently, lactic acid bacteria are commonly used as fermentation strains. Currently, approximately 20 strains of lactic acid bacteria have been successfully applied to plant-based fermentation, with *Lactiplantibacillus plantarum*,

*Lactobacillus acidophilus*, *Leuconostoc lactis*, and *Levilactobacillus brevis* among the most prevalent [9]. In particular, lactic acid bacteria fermentation of plant-based materials to produce functional food is one of the inevitable development trends of food in the future [10].

Currently, the categories of fermented plant-based products with lactic acid bacteria include fruits, vegetables, tea, and grains. The different types of fermented plant products are detailed in Table 1. However, a significant number of plant-based by-products are generated each year from fruits, vegetables, and grains, which contain a large amount of nutrients and active ingredients. Fermenting some plant-based by-products with lactic acid bacteria can produce plant-based beverages, livestock feed, etc. During this fermentation process, the nutrients and active substances are degraded by lactic acid bacteria into forms that are easier to absorb, which also degrades toxic substances and masks undesirable flavors. For example, Cerasus humilis kernels (CHK) have high nutritional value, but the deep processing of CHK is limited by a high content of amygdalin. Intracellular enzymes of lactic acid bacteria can degrade amygdalin, providing a new theoretical basis for enhancing nutrient utilization in CHK and promoting deep processing and comprehensive utilization technology of CHK [11]. Kimoto-Nira et al. produced a functional product from fermented shiikuwasha pomace by lactic acid bacteria. Shiikuwasha pomace is rich in glucose, fructose, etc. *L. plantarum* E58 can utilize these carbon compounds to produce lactate and acetate. By analyzing fermented pomace with a taste-sensing system, it was found that this process effectively masks the anionic bitter aftertaste in the fruit pomace. Furthermore, compared to unfermented pomace, the active substance nobiletin was protected, and its content was not reduced during the fermentation process. This utilization of nobiletin gives the fermented pomace product certain functional and practical properties [12]. Apple pomace is one of the rich and inexpensive sources of phenolic compounds. β-glucosidase and decarboxylase produced by lactic acid bacteria can release compounds bound in complex cell wall structures in the pomace and convert phenolic compounds into other substances. This process leads to changes in the types and content of phenolic compounds in fermented pomace. For example, the total phenol content in the pomace fermented by *L. plantarum* KKP 1527 increased by about 30% compared to the original, and the content of gallic acid, procyanidin A2, protocatechuic acid, and procyanidin B2 increased [13]. Lye et al. employed durio zibethinus, artocarpus champeden, and garcinia mangostanan by-products as immobilized substrates for lactic acid bacteria (*L. acidophilus* FTDC 1331, *L. acidophilus* FTDC 2631, *L. acidophilus* FTDC 2333, *L. acidophilus* FTDC 1733, and *Lactobacillus delbrueckii* subsp. lactis FTCC 0411) fermentation, resulting in an improved protein hydrolysis activity of fermented soymilk and enhanced ACE enzyme activity [14].

**Table 1.** Types of lactic-acid-bacteria-fermented plant-based products.

| Categorization | Offerings | Characteristic | Bibliography |
|---|---|---|---|
| Fruit and vegetable products | Fermentation of strawberry juice by *L. plantarum* and *L. acidophilus* | Enhance the color of anthocyanins to give the juice an orange color and increase the antioxidant capacity of the beverage | [15] |
| | Lactic acid bacteria mixed fermentation fermented Chinese bayberry pomace | Protects the color of the drink and slows down the degradation of anthocyanins. | [16] |
| | Low-sugar beverage with fermentation of *L. plantarum* of red jujube fruits and bamboo shoots | High nutritional value, high antioxidant capacity, and high dietary fiber content | [17] |
| | Fermentation of blueberry pomace by a mixture of *Lacticaseibacillus rhamnosus* and *L. plantarum* | Increased active substances to improve cholesterol-lowering ability | [18] |
| | *Limosilactobacillus reuteri*-fermented apple juice | Enhanced antioxidant capacity and flavors of fermented juices | [19] |

**Table 1.** *Cont.*

| Categorization | Offerings | Characteristic | Bibliography |
|---|---|---|---|
| grain | *Lentilactobacillus kisonensis*-fermented black barley | Modulation of high-fat-diet-induced dysbiosis of intestinal flora | [20] |
| | *Leuconostoc mesenteroides* subsp. *mesenteroides*- and *L. plantarum*-fermented various beans and rice yogurt | Improved flavors, sweetness, acidity, and texture of yogurt | [21] |
| | Fermented soybean–corn mixture of *L. plantarum*-, *Pediococcus acidilactici*-, and *Leuc. mesenteroides*-fermented soybean–corn mixture | Increased free amino acid content and enhanced sweetness | [22] |
| tea | *L. paracasei* subsp. *paracasei*-fermented green tea and fishweed | Inhibits the production of fat, with anti-obesity effects | [23] |
| | fermented tea | High anticholinesterase and anti-angiotensin-converting enzyme activity | [24] |
| | *Streptococcus thermophilus*- and *L. plantarum*-fermented black tea | Has therapeutic potential to improve antioxidant defenses and protect organisms from oxidative damage | [25] |

## 3. Advancements in Lactic Acid Bacteria Fermentation of Plant-Based Products

In the early stages, the application of lactic acid bacteria fermentation in plant-based products was mainly concentrated in fruits, vegetables, and tea. Compared to fermented dairy products with lactic acid bacteria, the research into and development of fermented plant-based products have been relatively delayed [26,27]. At present, the fermentation method of plant products is to add lactic acid bacteria fermentation liquid or freeze-dried lactic acid bacteria powder to the plant-based material for fermentation. In recent years, the process of lactic acid bacteria fermentation of plant-based products has been continuously developing. For example, the continuous screening of excellent strains specialized for the fermentation of plant-based products, the collaborative fermentation of multiple strains, etc. These advancements provide a solid technical foundation for enhancing the flavors, nutrient content, and health benefits of plant-based fermented products. The process of fermenting plant-based products by lactic acid bacteria is described in Figure 2.

Compared to traditional direct-pitch fermentation, immobilized bacteria fermentation is conducive to controlling the fermentation and improving the yield [28]. Immobilized bacteria fermentation is the use of immobilized plant-based by-products as a carrier for lactic acid bacteria fermentation, which not only solves the problem of pollution to the environment caused by plant-based by-products but also promotes the reuse of the active ingredients in the plant-based material by lactic acid bacteria. For example, *Oenococcus oeni* can improve malolactic fermentation in wine by using grape skins and stems as immobilized materials, and these natural products not only solve the biggest problem of solid by-products in the wine industry, but also have a minimal negative impact on the final product [29]. Immobilized lactic acid bacteria fermentation not only reasonably solves the problem of waste of plant by-products resources, but also ensures the survival of lactic acid bacteria to a certain extent.

Compared to the single strain fermentation of lactic acid bacteria, a synergistic fermentation of multiple strains can significantly improve the antioxidant property [30], bacteriostatic capacity, etc., of plant foods, as well as produce more metabolites and form richer flavors [31]. For instance, Li et al. proved that different combinations of lactic acid bacteria (*L. acidophilus* 85, *Lacticaseibacillus casei* 37, *Lactobacillus helveticus* 76, and *L. plantarum* 90) could increase vitamin content in fermented jujube juice [32]. In a similar vein, Gardner et al. found that the inoculation of *L. plantarum*, *P. acidilactici*, and *Leuc. mesenteroides* in mixed vegetable juices effectively improved the flavor, texture, safety, and storage resistance of the juices [22].

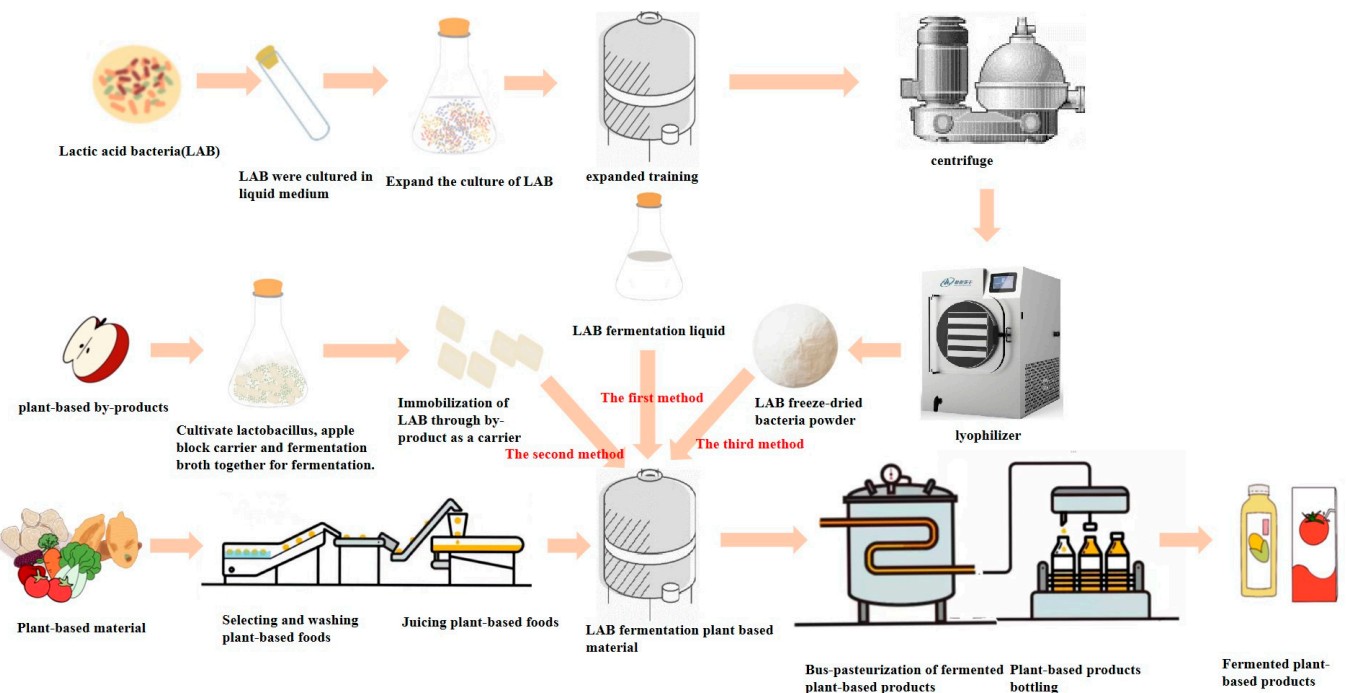

**Figure 2.** The process of fermentation of plant-based products by lactic acid bacteria.

## 4. Effect of Lactic Acid Bacteria Fermentation on Active Ingredients in Plant Foods

### 4.1. Amino Acids

Lactic acid bacteria cannot synthesize all the essential amino acids needed for their growth and metabolism. However, lactic acid bacteria secrete proteases and peptidases that can break down proteins into amino acids and peptides [33]. Fermented foods often contain high levels of biogenic amines, which are formed by amino acids via decarboxylation by lactic acid bacteria and represent one of the main safety risks in fermented products. However, not all lactic acid bacteria are the primary producers of biogenic amines in fermented foods. An increasing number of researchers have been screening lactic acid bacteria from kimchi that have the ability to degrade biogenic amines [34,35]. The ability of lactic acid bacteria to degrade biogenic amines is due to the multicopper oxidase (MCO) genes, which encode the enzymes responsible for biogenic amines degradation. Lactic acid bacteria secrete enzymes that degrade biological amines (such as MCOs and amine oxidases), which can degrade biological amines into aldehydes and avoid the accumulation of biological amines. For example, Lee et al. screened lactic acid bacteria (*L. brevis* PK08, *Lactiplantibacillus pentosus* PK05) from kimchi that significantly degraded bioamines [36]. Lactic acid bacteria capable of degrading bioamines not only solve the problem of bioamines production from the source, but also improve the safety of lactic acid bacteria as a starter culture. Because amino acids can exhibit acidic, sweet, and bitter characteristics, they play an important role in the flavor characteristics of fermented plant-based products [37]. Lactic acid bacteria have the capacity to metabolize soluble substances in the plant-based material, such as certain fats and vitamins, into amino acids. Additionally, lactic acid bacteria can utilize or transform bitter amino acids, enhance or increase the aromatic amino acid content, and thus contribute to the development of unique flavors in fermented plant-based products. When lactic acid bacteria ferment a mixture of soy and corn, enzymes secreted by the lactic acid bacteria hydrolyze the proteins, thereby increasing the concentration of free amino acids. Sensory evaluations have shown that this increase in sweetness after fermentation is associated with an increase in the content of sweet amino acids (Gly, Ala, and Lys) [38]. Furthermore, Chen et al. demonstrated that *L. rhamnosus* and *Gluconacetobacter xylinus* ATCC23767 fermentation can influence the levels of flavor-presenting amino acids in fruit juices, reducing the concentration of bitter amino acids by over 85% in Yacon–Litchi–

Longan juice [39]. Glutamate is an aroma enhancer that can improve the flavor of foods and beverages made from plants. Research by Sawada et al. indicated that the fermentation of rice kimchi under warm conditions significantly increases the glutamic acid content [40]. γ-Aminobutyric acid, classified as a functional non-protein amino acid, possesses important biological functions that include mediating hormone secretion, improving lipid metabolism, and reducing blood pressure [41]. Although plant-based material contains almost no γ-aminobutyric acid, numerous studies have shown that lactic-acid-bacteria-fermented products can produce abundant amounts of γ-aminobutyric acid [42]. This is related to the conversion of L-glutamic acid to gamma-aminobutyric acid by lactic acid bacteria during fermentation. Examples of this conversion also include grape [43] and kiwifruit [44], where the concentration of γ-aminobutyric acid in fermentation broth is increased. In the case of date residue, which is nutritious but underutilized in the production process, *L. brevis* JCM 1059T and JCM 1061 can degrade the monosodium glutamate and convert glutamic acid to γ-aminobutyric acid with a conversion rate of 80–90% [45]. Studies have shown that the fermentation of plant-based products with lactic acid bacteria can enhance the nutritional value and flavor profile of products by increasing the content and variety of amino acids.

*4.2. Vitamins*

Plant-based products are naturally rich sources of vitamins [46], but often have low levels of vitamin B and vitamin K. Lactic acid bacteria fermentation not only creates an acidic environment that enhances the stability of vitamins in plant-based products, but also facilitates vitamin synthesis [47]. Vitamin K plays an important role in blood coagulation and bone health in humans. But there are limitations in the extraction and synthesis methods of vitamin K in the actual production process, such as the existence of low product yields and environmental problems. Lactic acid bacteria fermentation can solve the above problems. Morishita et al. used lactic acid bacteria with a high yield of vitamin K to ferment soy milk, which increased the vitamin content in soy milk and could supplement the vitamin K needed by the human body [48]. Vitamin B deficiency can impact intestinal health, morphology, and inflammation, as well as contribute to the development of intestinal diseases [49,50]. Studies have shown that fermentation by lactic acid bacteria can increase the concentration of riboflavin and folic acid in food. Additionally, the combination of passion fruit by-products and *S. thermophilus* in fermented soymilk promotes the production of folic acid [51]. Furthermore, high concentrations of folic acid are synthesized during the fermentation of amaranth sourdough by *L. plantarum* CRL 2106, CRL 2107, and *Leuc. mesenteroides* subsp. *mesenteroides* CRL 2131 [52]. Li et al. also demonstrated that the fermentation of jujube juice by *L. acidophilus* 85, *L. casei* 37, *L. helveticus* 76, and *L. plantarum* 90 increased the content of vitamin P [32]. Vitamin P has antioxidant and anti-inflammatory properties, while β-carotene offers various benefits to the body, such as regulating lipid metabolism and reducing the inflammation caused by oxidative stress [53]. In the case of β-carotene, which cannot be synthesized by the human body and must be obtained through diet, *Pediococcus pentosaceus* or *Latilactobacillus sakei* fermentation of tomato pulp resulted in a more than four-fold increase in β-carotene content [54]. Vitamin C, the most abundant water-soluble vitamin in orange juice, provides strong antioxidant effects and nutritional value for human health. The fermentation of orange juice by *L. plantarum*, *L. fermentum*, *L. acidophilus*, and *Bifidobacterium longum* subsp. *longum* increased the content of vitamin C by 19.42%, 16.72%, 16.25%, and 6.80%, respectively, because vitamin C can be synthesized during the fermentation process by lactic acid bacteria [55]. Additionally, Cai et al. demonstrated that the fermentation of kiwifruit extract by *L. paracasei* LG0260 significantly increased the concentration of vitamin C [44]. Vitamin B12 is essential for overall health as it prevents pernicious anemia by promoting the development and maturation of red blood cells, as well as promoting the metabolism of carbohydrates, fats, and proteins [56]. Research by Palachum et al. indicated that *L. plantarum* ferments guava pulp to produce guava gummy jelly products with high vitamin B12 content [57].

### 4.3. Volatile Compounds

Lactic acid bacteria can affect the release and degradation of volatile compounds such as aldehydes, alcohols, and acids in plant products, resulting in changes in the content and distribution of volatile compounds in plant-based products. Grapes and citrus fruits are rich in proteins and amino acids, which can be broken down by enzymes present within lactic acid bacteria such as dehydrogenases, decarboxylases, and transaminases. This breakdown process leads to the production of aldehydes, alcohols, and acids. In Wu's study, the total volatile content in fermented grape juice increased by 65.75%, demonstrating that lactic acid bacteria fermentation can enrich the volatile aroma compounds in grape juice. Moreover, nerol, which was identified as the unique volatile in Muscat grapes, was found to be increased by 95.38% after fermentation. Myrcene is characterized by a pleasant and subtle resinous aroma. The study also found that unfermented grape juice contains six alkenes, whereas only myrcene is detected after fermentation, with a concentration 2.23 times higher than before fermentation. This process may be related to the hydrolysis of alkenes by β-glucosidase of lactic acid bacteria in grape juice [58]. Quan et al. fermented orange juice with six strains of lactic acid bacteria (*L. plantarum*, *L. fermentum*, *L. acidophilus*, *L. rhamnosus*, *L. paracasei*, and *B. longum*). The study found that 18 new volatile compounds were added, which may be due to the synthesis of specific terpenes by certain lactic acid bacteria. At the same time, the total alcohol content (e.g., (-)-4-pinene and linalool) in the fermented juice increased by 1.06–8.03 times compared to the control group, which may be related to the decomposition and metabolism of glucose and amino acids in the juice by lactic acid bacteria. The Odor Activity Value (OAV) reflects the "sensory intensity" of a certain aroma substance. The larger the OAV, the greater the contribution of ingredients to the flavor of the food. It is known from the analysis combining sensory evaluation and aroma-active volatile profiles that 22 aroma-active compounds were detected in the sample fermented by *L. paracasei.* Moreover, this sample was found to be the most popular in sensory evaluation, exhibiting a pleasant fermented flavor, orange aroma, and sourness. Among these compounds, 3-carene (OAV 152), pentanal (OAV 11), and B-Caryophyllene (OAV 3) contributed to the lemon, woody, and almond flavors [55]. The variation in the content of alcohols and ketones substances is conducive to increasing the volatility of product flavors and the complexity of aromas [59]. The content of ketone substances in mango juice increases after fermentation by lactic acid bacteria (*L. plantarum* NCU116, *L. acidophilus* NCU402, and *L. casei* NCU215), particularly a significant increase in the content of p-Isopropenyl toluene, enriching the ideal aroma of mango in terms of fruity and sweet flavors. Terpenoids and aldehyde substances have adverse effects on the flavor of fermented products, such as limonene, cyperene, and (+)-alpha-amorphene, which may exhibit slight irritating odors. However, these volatile substances are reduced after fermentation. 2-hexenal produces a strong green odor. The content of 2-hexenal in mango juice is reduced after fermentation, and this process may be related to the conversion of aldehyde substances into alcohol or acid substances by *L. plantarum* NCU116 [60]. Esters can offer sweet and fruity odors of juice [61]. Peach pulp was fermented by five commercial probiotics (*L. plantarum* 21802, *L. brevis* 6239, *L. pentosus* 21798, *L. alimentarius* 21852, and *L. pentosus* 21832) by Yang et al. It was found that γ-Hexalactone, which provides sweet and fruity flavors, increased by 82.47%. During the fermentation process of lactic acid bacteria, various new ketone substances are produced, such as β-Damascenone, which has the highest OVA value. It is considered a major characteristic flavor compound in fermented peach pulp and contributes to the creation of a sweet and strong floral flavor [62].

### 4.4. Extracellular Polysaccharides

Extracellular polysaccharides (EPS) are synthesized by microorganisms during carbohydrate fermentation and contribute to the development of food flavors [63]. Lactic acid bacteria can use a variety of biosynthetic pathways to produce a wide range of EPS. EPS have considerable potential for the development of functional foods due to their antioxidant, anticancer, and antibacterial properties [64]. Additionally, EPS exhibits excellent

water retention and rheological properties [65], making them an ideal natural food additive. As a result, EPS are widely used as stabilizers, emulsifiers, and gelling agents to enhance the stability and rheology of foods and beverages. Lactic acid bacteria that produce EPS include *L. lactis*, *B. longum*, and *L. lactis* [66]. Currently, lactic acid bacteria that can produce EPS are used to produce dairy products such as yoghurt and cheese [67]. The fermentation of plant-based beverages by lactic acid bacteria not only increases the content of EPS, but also improves the texture and consistency of the fermented beverages [68]. For example, Hickisch et al. use milk to produce plant-based yogurt, and the production of EPS played a crucial role in improving the texture of yogurt and increasing gel hardness [69,70]. Yang et al. demonstrated that the addition of EPS significantly influenced the viscosity, water-holding capacity, hardness, and microstructure of yogurt [71]. The study demonstrated that EPS released by lactic acid bacteria in wine and other fermented beverages interact with tannins and salivary proteins, leading to a reduction in the astringency of the wine, and this EPS is β-glucan [72]. This alteration in astringency and continuity indirectly enhances the aroma of the product, providing evidence that EPS produced by lactic acid bacteria contributes to fruit juice improvement. Furthermore, Chen et al. confirmed that lactic acid bacteria fermentation had an impact on the EPS content of Yacon–Lychee–Longan juice. Compared with the unfermented juice, the EPS content after fermentation could reach 67.02 mg/L, which was twice the original [39]. Whole wheat fresh noodles were fermented with *Weissella cibaria* producing EPS by Du et al. The organic acids produced during the fermentation process have a negative impact on the viscoelasticity of the acid dough, but the accumulation of EPS reduces the acidification of the acid dough, prevents the depolymerization of the gluten macropolymer, and improves the elasticity of the fermented dough [73].

*4.5. Organic Acids*

Lactic acid bacteria fermentation can promote the conversion and utilization of organic acids in plant materials [74]. This process results in milder flavors and improved sensory properties, nutritional qualities, and other characteristics of the fermented plant-based products [75]. Many plant-based products contain a lot of tartaric acid and malic acid with a strong sour flavor. Lactic acid bacteria cannot only degrade the organic acids mentioned above, but also convert them into lactic acid, giving the fermentation products a pleasant sour taste. In unfermented grape juice, the main organic acid is tartaric acid [45], which leads to high acidity in wine due to the presence of L-malic acid. However, the fermentation by lactic acid bacteria can convert L-malic acid to lactic acid, which has milder flavors. Consequently, the fermented juice becomes softer than unfermented wines in taste, reduces the wine's acidity, and enhances the flavor characteristics of grape juice. *Sea buckthorn*, known for its sour taste due to its high organic acid content, can be fermented by *L. plantarum*, *L. plantarum* subsp. *argentoratensis*, and *O. oeni* to produce high amounts of lactic acid. This fermentation process also reduces the formation of bitter tasting compounds like quinine [76]. Apple juice has a strong sour taste because it contains a large amount of malic acid, and lactic-acid-bacteria-fermented apple juice can convert fumaric acid into succinic acid under anaerobic conditions, alleviating the sour taste of the juice [19]. Yang et al. fermented chestnut with *L. casei* CU and *L. fermentum* KF5. The results showed that the levels of organic acids, including 2-hydroxycaproic acid, 4-hydroxybutyric acid, succinic acid, and erythronic acid, were found to be higher in fermented chestnut compared to unfermented chestnut. This process is due to the degradation of macromolecular substances in chestnut by lactic acid bacteria [77]. The organic acid produced by lactic acid bacteria not only has the function of enhancing product flavor, but also has the function of inhibiting bacteria. Researchers have discovered the antifungal activity of fermented tomatoes, among which organic acids have been identified as antifungal compounds [78]. According to Qiao et al., sugarcane juice fermented by *Lactobacillus* HNK10 and *Lactococcus* HNK21 had a strong lactic acid flavor and demonstrated strong antibacterial activity by inhibiting the growth of undesirable bacteria [7].

### 4.6. Phenolic Compounds

Phenolic compounds are essential bioactive substances found in plant-based products such as grapes and strawberries. They have a positive impact on human health, physiological regulation, and disease resistance [79]. However, the short shelf-life of fresh plant-based products often leads to an underutilization of phenolics. Lactic acid bacteria fermentation not only prolongs the shelf-life of plant-based products, but also increases the quantity and variety of phenolics present in them. Curiel et al. showed that *L. plantarum* C2 fermentation made the total phenol concentration 5-fold higher than non-fermented myrtle berries homogenate, with the concentration of anthocyanins increasing 10-fold, which is inseparable from plant-based material biological acidification [80]. Similarly, Chen et al. observed an increase in phenolics, including total phenolic acids and total flavonoids, in kiwifruit pulp fermented by three different lactic acid bacteria species (*L. acidophilus*, *L. plantarum*, *L. casei*), compared to nonfermented kiwifruit pulp. This change may be due to the microbial enzymes produced by lactic acid bacteria [81]. Some studies have shown that phenolics are associated with the functional properties of grape juice, such as anti-inflammatory and antioxidant activities. During the early stage of grape fermentation by *L. plantarum* 21802 and *L. brevis* 6239, the content of all detected phenolic compounds significantly increases, indicating that lactic acid bacteria fermentation enhances the release of phenolic compounds from grape pulp. In the later stage of fermentation, lactic acid bacteria biotransform caffeic acid and $\alpha$-arbutin [58]. Kwaw et al. found that, after the fermentation of mulberry juice by different lactic acid bacteria, the total phenolic acid concentration increased by about 1.5 times, and the ferulic acid concentration increased by 2.8 times after fermentation by *L. plantarum*, which was related to the conversion of complex phenolic substances in mulberry by lactic acid bacteria into free forms [82]. The influence of fermentation by lactic acid bacteria (*L. plantarum*, *L. rhamnosus*) on phenolic compounds in lychee was investigated by Tang et al. After lactic acid bacteria fermentation, the content of Quercetin-3-O-rutinose-7-O-$\alpha$-L-rhamno-side (QRR) decreased, while gallic acid, 4-hydroxybenzoic acid, proanthocyanidin B2, catechin, and quercetin contents were higher compared to before fermentation. This may be related to the metabolic pathway of 3-O-rutinose-7-O-$\alpha$-L-rhamnoside, which appears to involve the direct fission of the C-ring at the C2-O1 and C3-C4 bonds. Short-chain fatty acids (SCFA) are significant metabolic products of intestinal microorganisms, and the level of SCFA partly indicates the growth status of the gut microbiota. The supernatant obtained after fermenting lychee was subjected to anaerobic fermentation with human intestinal flora extract. The results showed that the production of SCFA was higher in samples fermented by either *L. plantarum* or *L. rhamnosus* individually, as well as in the co-fermentation samples, compared to the unfermented group. This indicates that the gut microbiota status was most favorable in the samples fermented by lactic acid bacteria. This may be attributed to the ability of lactic acid bacteria to convert phenolic compounds in lychee into smaller molecules, which is more favorable for the metabolism and transformation of the gut microbiota [83]. Compared with unfermented papaya puree, *Leuconostoc pseudomesenteroides* 56 fermented papaya puree had the highest concentrations of protocatechuate, coumaric acid, and ferulic acid after 7 days of storage at 4 °C, and the inhibition of intestinal $\alpha$-glucosidase activity by phenolic compounds is considered to serve as a mechanism to exert an antidiabetic effect. In this study, protocatechuic acid, caffeic acid, coumalic acid, and ferulic acid were strongly positively correlated with $\alpha$-glucosidase inhibitory activity [84].

## 5. Effect of Lactic Acid Bacteria Fermentation of Plant-Based Products on Functional Properties

### 5.1. Improvements in Flavor

Plant-based materials' lipids, proteins, and carbs can all be broken down into flavor precursors by lactic acid bacteria's enzymes, which are then converted into aromatic compounds. The flavors found in the lactic acid bacteria fermentation of plant-based beverages mainly stem from organic acids, amino acids, sugars, and volatile flavor components. Certain plant-based beverages, such as celery and bitter melon, often possess pungent odors

and lack satisfactory flavors. However, when celery and bitter melon are fermented with lactic acid bacteria, their flavors are significantly improved. Some plant-based beverages made from legumes often possess bitter and beany flavors. However, during the fermentation process of lactic acid bacteria, substances such as fat and protein in legume products are decomposed, eliminating unpleasant odor substances and generating metabolites like diacetyl, ethyl acetate, and acetaldehyde, imparting aroma and acidity to the products [85]. The impact of lactic acid bacteria fermentation on the flavor of pea protein isolate (PPI, from pisum sativum) was investigated by Arteaga et al. The research findings indicated that 24 h of fermentation with *L. perolens* resulted in the highest buttery aroma in PPI, which could be attributed to the metabolism of *L. perolens* [86]. Legume-based water extracts were fermented by Demarinis et al., and it was found, through sensory evaluation analysis, that the sensory characteristics of the fermented legume-based extract beverages are associated with the production of lactic acid. The original bitterness of the legume-based extract beverages was masked by milk flavor or vegetable notes, resulting in a reduction in unacceptable sensory characteristics [87]. Wu et al. made bread from a sourdough fermented by lactic acid bacteria and corn oil. Their study found that the difference in VOCs of sourdough contributes to the unique flavor of the bread. Additionally, using the GC-O-MS method, 15 odor-active VOCs were identified in the bread samples made from fermented sourdough. For example, (E, E)-2,4-decenedial, 2-amylfuran, 1-octene-3-ol, 3-methylthio-1-propanol, and (E)-2-nonenedial are key aroma compounds with high flavor dilution coefficients and odor activity values (OAVs), distinguishing them from other bread varieties. Furthermore, sensory evaluation analysis indicated that the bread made from the combination of lactic acid bacteria and corn oil in the fermentation process received the highest scores compared to non-fermented dough [88]. The rose and shiitake mushroom mixed beverage was fermented with five strains of lactic acid bacteria by Qiu et al. Free amino acids and flavor nucleotides were identified as the main contributors to the flavor of the mixed rose and shiitake mushroom beverage. It was found that lactic acid bacteria fermentation significantly increased the content of glycine and serine in the beverage. At the same time, the content of flavor nucleotides (such as 5′-nucleotides) in the fermented beverage was also increased, which was closely related to the nucleosidase secreted by lactic acid bacteria. Furthermore, the results of electronic tongue analysis demonstrated that lactic acid bacteria fermentation effectively reduced the signals of bitterness, astringency, aftertaste-A, and aftertaste-B, leading to an overall improvement in the taste profiles of the fermented samples [89]. Moreover, Ginseng, lotus leaf, poria cocos, rice bean, tangerine peel, and cassia are mixed and crushed in specific proportions. The mixture is then blended with hot water and centrifuged to obtain the supernatant, which is known as the FH 06 beverage. The flavor of FH06 beverages is significantly improved after probiotic fermentation by lactic acid bacteria (*L. fermentum* grx 08, *L. rhamnosus* hsryfm 1301, *L. rhamnosus* grx 10, *L. plantarum* 67, and *L. plantarum* S7). Volatile compounds with an odor activity value (OAV) greater than 1 are generally considered to make significant contributions to the overall aroma. Cinnamaldehyde is the main flavor compound in FH06 beverages. It has a distinct tangerine peel flavor before fermentation, which is usually not easily accepted in drinks. However, after fermentation by lactic acid bacteria, the content of cinnamaldehyde decreased from 3402.57 μg/L to 7.55 μg/L, and the OAV value dropped from 4.54 to 0.01, indicating that it was no longer a key compound in the flavor of the fermented beverage. Finally, the GCMS and sensory analysis of FH06 fermented by the GRX08 strain showed that fermentation removed the original grassy, tangerine peel, and bitter tastes, highlighting the fruity aroma and mild acidity, making the fermented FH06 more easily accepted by consumers [90].

### 5.2. Antioxidant Effects

The antioxidant function is a significant aspect of the probiotic function exhibited by lactic acid bacteria during the fermentation of plant-based products. This enhancement in antioxidant function is mainly attributed to the increase in phenolics and vit-amins, among

other active compounds, in the fermented plant-based products [91]. In addition, compared to unfermented products, an environment is created by the organic acids produced during lactic acid bacteria fermentation that protects antioxidant activity substances and stabilizes antioxidants. The antioxidant capacity of fermented pear juice positively correlates with the content of phenolics such as vanillic acid and arbutin [92]. During apple juice fermentation, *L. plantarum* ATCC14917 consumes available glucose molecules in phenolic compounds and generates metabolic substances such as O-caffeoylquinic acid, quercetin, and phloretin. These metabolites possess either more hydroxyl groups or lower steric hindrance, resulting in an improvement in the antioxidant capacity of fermented apple juice [93]. The impact of *L. plantarum*, *L. acidophilus*, and *L. paracasei* on the antioxidant activity of mulberry juice was studied by Kwaw et al. It was discovered that, during fermentation, lactic acid bacteria can deglycosylate more glycosylated phenolic compounds in mulberry juice, leading to the release of more soluble conjugated or insoluble bound phenolic compounds from plant cell walls. This ultimately increases the antioxidant activity of fermented mulberry juice [82]. Lactic acid bacteria can grow and ferment in grape juice, resulting in fermented grape jam and its by-products with antioxidant activity. In a study conducted by Wu et al., mice were fed with fermented grape juice, separately fermented by *L. plantarum* 21802 and *L. brevis* 6239. The evaluation of SOD activity and MDA level was evaluated to determine the antioxidant capacity of the liver and serum. The results showed that caffeic acid and alpha-arbutin were not detected in unfermented grape juice, but were present in the late or terminal stages of lactic acid bacteria fermentation, indicating that other phenolic substances were bioconverted into these compounds at these stages, resulting in increased the antioxidant capacity of fermented grape juice [58]. Additionally, the fermentation of portulaca oleracea L. juice by lactic acid bacteria (*L. plantarum* POM1, T1.3, and EnFIII3, *L. brevis* POM4, Furfurilactobacillus rossiae 2MR8, *P. pentosaceus* CILSWE5, *Leuc. mesenteroides* OP9, *Api-lactobacillus kunkeei* B7) significantly increases the total antioxidant capacity of the fermentation broth. This process effectively preserves the original levels of vitamins C, A, and E and even increases the levels of vitamin B2 and phenolics [5]. The above research indicates that lactic acid bacteria fermentation not only alters phenolic compounds, but can also achieve the purpose of enhancing antioxidant capacity by changing the content and types of vitamins.

*5.3. Antimicrobial Properties*

The antimicrobial effect of lactic acid bacteria is associated with the production of various active metabolites during fermentation, such as organic acids, hydrogen peroxide, diacetyl, carbon dioxide, fatty acids, bacteriocins, biosurfactants, etc. [94]. Lactic acid bacteria can use the citric acid in lemon as carbonic acid to produce a large amount of lactic acid, which can increase the antibacterial ability of fermented lemon juice. *L.plantarum* LS5 exhibits enhanced antimicrobial properties during the fermentation of lemon juice compared to unfermented lemon juice, with inhibitory effects against both Salmonella typhimurium and *Escherichia coli* O157:H7 [95]. A study conducted by Zhong et al. further demonstrated that the enhanced antibacterial activity in fermented blueberry juice was associated with a decrease in pH and an increase in organic acid content [96]. Omedi et al. studied the antifungal activity of lactic acid bacteria (*L. plantarum*, *L. pentosus*, and *P. pentosaceus*) against fungal strains and found that lactic acid bacteria showed strong antifungal activity against *Aspergillus niger*, *Cladosporium sphaerospermum*, and *Penicillium chrysogenum*. This activity is attributed to the phenolic acid metabolites released during the fermentation process. Phenolic acids act as antifungal compounds that can act independently or synergistically to inhibit fungal growth [97]. Luz et al. obtained water-soluble extracts (WSEs) by centrifugation after fermenting sour dough with lactic acid bacteria. These WSEs were lyophilized and were used for antifungal activity tests. The extracts produced by *L. plantarum* CECT 749 and *L. bulgaricus* CECT 4005 were effective against strains of *Fusarium* spp., *Penicillium* spp., and *Aspergillus* spp. LC-ESI- MS-TOF analysis of WSEs revealed that phenolic acids such as gallic acid, chlorogenic acid, caffeic acid, and

syringic acid were the substances that acted as antibacterial agents [98]. Kim et al. studied the biotransformation of *L. breves* DF01 and *P. acidilactici* K10 on mulberry fruit extract. The results showed that lactic acid bacteria could reduce the growth and biofilm formation of Salmonella typhimurium and make mulberry fruit extract have antibacterial activity [99].

*5.4. Anti-Inflammatory Function*

Numerous studies have demonstrated the anti-inflammatory effects of lactic acid bacteria in fermented foods [100]. TNF-α, IL-6, and IL-1β are inflammatory cytokines that reflect the level of immune response in the body. Lactic acid bacteria fermentation can confer anti-inflammatory activity to fermented plant-based products by increasing or transforming the active ingredients. Curcumin is the main bioactive component of turmeric. Compared with unfermented turmeric, fermentation with *L. fermentum* significantly increased the curcumin content by 9.76%. The fermented turmeric suppresses the expression of proapoptotic tumor necrosis factor-alpha (TNF-α) and Toll-like receptor-4 (TLR4) in lipopolysaccharide-induced RAW 264.7 cells. In addition, the researchers found that the anti-inflammatory activity of the fermented turmeric was exerted through suppression of the c-Jun N-terminal kinase (JNK) signal pathway [101]. *Scrophularia buergeriana* has a variety of active ingredients. Pham et al. found that a sample of lactic acid bacteria fermentation of *Scrophularia buergeriana* extract demonstrated anti-inflammatory effects in lipopolysaccharide-treated RAW264.7 cells, inhibiting the production of nitric oxide (NO) and reducing the expression of RNA for INOS, IL-1β, IL-6, TNF-α, and COX-2. The results indicated the potential anti-inflammatory activity of lactic acid bacteria fermentation of *Scrophularia buergeriana* extract [102]. Sun et al. investigated the anti-inflammatory effect of soymilk fermented by *L. plantarum* and found that it significantly inhibited the release of inflammatory factors in the serum of mice after tube-feeding. This further demonstrated the ability of fermented soymilk to protect mice from colitis by inhibiting the release of inflammatory cytokines [103]. Kim et al. investigated the anti-inflammatory effect of lactic acid bacteria on biotransformed mulberry fruit extract. The study found that lactic acid bacteria inhibited the production of IL-8 in human intestinal epithelial cells induced by *S. typhimurium*. These results suggest that mulberry fruit extracts biotransformed by *L. brevis* DF01 and *P. acidilactici* K10 can partially regulate intestinal inflammation [99]. The NF-κB signaling pathway is involved in inflammation by stimulating the activation of viruses, tumor necrosis factor, and other factors [104,105]. Flavonoids and phenolic acids were classified as aglycone-polyphenols, and NF-B activation was inhibited by these compounds. Compared with unfermented tomato meat residues, lactic acid bacteria fermentation retains the biological activity of aglycone-polyphenols, which makes fermented tomato meat residues possess a 78% anti-inflammatory activity [106].

*5.5. Hypoglycemic Function*

α-Glucosidase and α-amylase are the two main glycosidases that participate in the metabolism of carbohydrates. Inhibitors of these two enzymes are considered an important medical treatment for diabetes. Fruits and vegetables contain natural α-amylase inhibitors, which inhibit the enzyme α-glucosidase, thereby delaying carbohydrate absorption and contributing to the management of hyperglycemia and diabetes complications. In a study by Sun et al., lactic acid bacteria (*L. casei* ATCC334, *L. plantarum* CICC20265, *L. acidophilus* CGMCC1.2686, *L. helveticus* CICC6024, and *L. paracasei* CICC20245) fermentation of pumpkin favored the inhibition of α-glucosidase and α-amylase activities. Among them, *L. helveticus*-fermented pumpkin had the lowest inhibition rate of α-glucosidase, which might be related to the biotransformation of phenolic compounds. The study suggests the hypoglycemic potential of fermented pumpkin juice [107]. Blueberries contain anthocyanins, phenols, organic acids, and vitamins, which have anti-obesity and antihyperglycemic activities. Studies have shown that the inhibitory effects of α-glucosidase and α-amylase are increased in *L. plantarum* J26-fermented blueberry juice (FBJ). This process is mainly related to the increase in phenolic substances in FBJ and the biotransformation of functional

substances by *L. plantarum* J26 [108]. Gao et al. used *L. plantarum* for the fermentation of Momordica charantia juice and found that fermented Momordica charantia juice was more effective than non-fermented Momordica charantia juice in reducing hyperglycemia, hyperinsulinemia, and hyperlipidemia in diabetic rats. Fermented Momordica charantia could be beneficial for ameliorating T2D by reducing the total carbohydrate content and improving the inhibition of α-glucosidase in vitro. This study demonstrated that lactic acid bacteria fermentation enhances the antidiabetic properties of bitter melon juice [109]. Mixed strain (*Lactiplantibacillus paraplantarum* CRL2051 and *L. plantarum* CRL2030) fermented pomegranate juice significantly reduced blood glucose in mice on a high-fat diet, which was associated with high concentrations of polyphenolic compounds in fermented pomegranate juice [110].

*5.6. Enhancement of Immune Function*

Lactic acid bacteria can enhance the ability of the body to fight against diseases by activating macrophages, promoting cell division, and producing antibodies [111]. Numerous studies have demonstrated that certain lactic acid bacteria have the capacity to inhibit colonic carcinoma, liver cancer, and lung cancer, both in vivo and in vitro [112]. Furthermore, during the fermentation process of food, certain lactic acid bacteria can enhance immune activity by secreting active substances with immunomodulatory functions [113]. The fermentation of herbs by lactic acid bacteria can increase the bioavailability and release of immune-active substances, such as polysaccharides, saponins, and polyphenols, which possess strong immunomodulatory activity. The oral intake of lactic-acid-bacteria-fermented herbs can promote intestinal secretion of immunomodulatory-active metabolites, such as EPS, short-chain fatty acids, and bacteriocins [114]. It was found that mice that ingested lychee juice fermented by *L. casei* could have an enhanced immunomodulatory activity. This effect is achieved by stimulating the spleen and thymus, as well as promoting the secretion of cytokines (IL-2, IL-6) and immunoglobulins (IgA, IgG, and SIgA) [115]. Nishioka et al. studied the effects of *Sparassis crispa* (SC) and lactic-acid-bacteria-fermented SC (SCL) on innate immunity. The SCL group significantly enhanced the accumulation of phagocytes, immunocytes, and C-C chemokine receptor-type-2- or phospho-Sky-expressing cells in the jejunum epithelial villi and spleen. The SCL group significantly enhanced the expression of genes involved in encoding various innate immune-related factors in a dose-dependent manner. In addition, the study found that the SCL group strengthened the phagocytosis of human monocytes against *Escherichia coli*, which is related to the β (1-3)-glucan in SC. The results showed that the oral administration of SCL significantly enhances innate immunity in mice and possibly humans [116]. Xeniji, a functional food made by fermenting various fruits and vegetables with lactic acid bacteria (*L. brevis*, *L. casei*, etc.), has been shown to activate T-lymphocytes and the cytotoxicity of natural killer (NK) and lymphokine-activated killer (LAK) cells in mice, thereby enhancing immunity [117].

*5.7. Regulation of Intestinal Flora*

When gut microbiota dysbiosis occurs, lactic acid bacteria with certain adhesive and colonization abilities can restore host resistance and the intestinal microbial barrier. Plant-based products fermented by lactic acid bacteria exhibits persistence in the human gastrointestinal tract and exerts positive effects on gut health through lactic acid bacteria and fermented metabolites [118]. Notably, lactic acid bacteria and *Akkermansia* are important genera of probiotic bacteria, known for their ability to produce substances that inhibit pathogens and combat metabolic diseases. Consequently, they play a crucial role in regulating the balance of gut microorganisms and improving host health. A study by Wen et al. revealed that *L. casei*-fermented lychee juice increased the relative abundance of *Firmicutes*, lactobacillus, and *Ackermansia* in the mouse intestinal microbiota, while reducing the abundance of *Bacteroidetes*, when compared to the normal group [115]. Zhu et al. investigated the effect of *L. plantarum* fermented black barley on NAFLD rats. Studies have shown that fermented black barley has a regulatory effect on the intestinal flora dysregulation

induced by a high-fat diet. This process is mainly due to the increase in flora diversity and the relative abundance of Bacteroidetes, the reduction in the *Firmicutes/Bacteroidetes* ratio, as well as the enrichment of some intestinal probiotics, such as Ackermania [20]. This fermentation process also promoted the secretion of mucosal SIgA, a critical mediator in the regulation of intestinal homeostasis, thereby protecting the mouse intestine and modulating the composition of the intestinal microbiota [114]. Additionally, short-chain fatty acids, which are non-direct nutrients produced by the gut microbiota, play important physiological regulatory roles. Cheng et al. demonstrated that the fermentation of blueberry pomace by *L. casei* had the potential to reinforce the intestinal barrier by increasing the production of short-chain fatty acids [119].

### 5.8. Anti-Tumor Function

Plant-based products are rich in various components that possess anti-cancer properties, such as vitamins, dietary fiber, and phenols [120]. Numerous epidemiological studies have consistently demonstrated a negative correlation between the consumption of plant-based products, such as fruits and vegetables, and the incidence of diverse chronic diseases, including cancer, stroke, and cardiovascular disease [121,122]. The fermentation of plant-based products by lactic acid bacteria can increase the production of active ingredients with anti-cancer functions, such as EPS, peptidoglycans, nucleic acids, and bacteriocins, conferring fermented foods with the ability to inhibit the growth of cancer cells [123]. Murthy et al. prepared a functional drink from pomegranate fermentation with lactic acid bacteria (*L. plantarum* VITES07 and *L. acidophilus* NCIM2903). It was found that fermented pomegranate juice was associated with the activity of tumor cell cleavage, which suggested that fermented pomegranate juice had anti-tumor potential. During fermentation, lactic dehydrogenase secreted by lactic acid bacteria can inhibit the formation and growth of tumor. In addition, the phenols of fermented pomegranate juice are also related to anti-tumor effects; for example, gallic acid has been shown to be cytotoxic to cancer cells [124]. Chaiyasut et al. demonstrated that a synbiotic formulation combining *L. plantarum* HII11 and inulin as functional foods offering protection against AOM-mediated colon cancer development in hosts [125]. SW480 CRC cells were treated with aqueous extracts of cherry silverberry fermented by mixed strains (*L. plantarum* and *L. casei*); compared with unfermented cherry silverberry extract and controls, fermented cherry silverberry extract exhibited the most potent tumor suppressor properties at 25–50 µg/mL. The anticancer properties of cherry silverberry extract were found to be related to the increased contents of epigallocatechin gallate, rutin, naringin, and quercetin after fermentation [126]. The main active ingredient in plant ginseng is saponins. However, natural saponins do not necessarily possess the most optimal molecular structure for physiological activity. In comparison to ordinary saponins, rare saponins exhibit specific anti-tumor effects. One or more enzymes produced by lactic acid bacteria can convert saponins into rare saponins, significantly increasing their content. Research by Xu et al. demonstrated that fermentation by *L. plantarum* enhances the conversion efficiency of diol saponins and increases the content of rare ginseng saponin-CK by 256%. This further confirms the ability of lactobacilli fermentation to enhance anti-tumor functionality through the bio-transformation of active substances [127].

### 5.9. Anti-Obesity Effects

Obesity is closely linked to metabolic syndromes such as dyslipidemia and diabetes. The prolonged consumption of a high-fat diet (HFD) is a significant risk factor for obesity. Moreover, lactic acid bacteria extracted from naturally fermented products have been found to possess anti-obesity effects, which positively influence weight control and help lower fat and lipid levels [128]. Utilizing the anti-obesity functional properties of lactic acid bacteria in the fermentation of plant-based sources can serve as a supplement to health functional products, and is particularly important in addressing diet-induced obesity. Lactic acid bacteria fermentation can enhance the active substances associated with anti-obesity effects,

thereby maximizing their potential through fermentation. For example, it has been found that apples and cabbage contain abundant polyphenols, or flavonoid compounds, which have anti-obesity effects. However, the absorption rate of some soluble polyphenols in the colon is low. The fermentation by lactic acid bacteria can convert the above compounds into forms that are more easily absorbed by the human body, thereby enhancing the anti-obesity effects of cabbage–apple juice. Researchers have demonstrated the anti-obesity effects and positive effects on lipid metabolism of fermented cabbage–apple juice by feeding it to obese mice [129]. As apple pomace contains a large number of polyphenols, adding polyphenols recovered from apple pomace to turbid apple juice can improve the utilization of plant-based by-products and maximize the application of polyphenols in apple juice through fermentation by lactic acid bacteria. Mixed lactic acid bacteria (*L. acidophilus* 6005, *L. plants* 21,805, and *L. fermentum* 21,828) ferment with the addition of polyphenols (FCAJP). Obesity is characterized by fat accumulation and higher blood lipid levels. The research findings revealed that the FCAJP showed potential to inhibit weight gain in mice, reduce fat accumulation, and regulate the blood lipid levels of obese mice by decreasing the ratio of the *Firmicutes/Bacteroidotas*, improving the Sobs, Ace, and Chao indexes of the gut microbiota and protecting intestinal tract health. This process may be related to the abundance of phenolic acids and organic acids in FCAJP after fermentation [130]. The vegetable juice was fermented using two lactic acid bacteria (*Companilactobacillus allii* WiKim39 and *L. lactis* WiKim0124). The study found that fermented vegetable juice could reduce weight gain and liver fat accumulation in mice. The results showed that indole-3-lactic acid, leucine, phenyllactic acid, and other metabolites in fermented vegetable juice had significant inhibitory effects on intracellular lipid accumulation [131].

*5.10. Improves Osteoporosis*

Osteoporosis is a systemic bone disease characterized by reduced bone mass and the degradation of the bone microarchitecture, consequently increasing bone fragility and the risk of fractures. While traditional anti-osteoporosis medications in the market are accompanied by various side effects, an increasing number of women are seeking botanical alternatives. Plant-based sources, such as *Cimicifuga racemosa* and red clover, contain phytoestrogenic substances that successfully treat women's postmenopausal osteoporosis [132]. Fermentation is believed to enhance the pharmacological effects of herbs. Lactic acid bacteria have demonstrated anti-osteoporotic potential and have been shown to increase mRNA expression of bone-metabolism-related markers OCN, OPG, ALP, BSP, and RUNX2 in osteoblasts, indicating their potential for preventing bone-loss-related diseases [133]. Miura et al. investigated the effects of a fermented soy product from lactic acid bacteria (PS-B1) on preventing the decrease in bone mineral content and deterioration of trabecular bone structure in ovariectomized mice. The results showed that PS-B1 has the effect of retarding the decrease in bone mineral density and the deterioration of the trabecular bone structure. The main component of PS-B1 is glutamic acid, which is the main reason why PS-B1 has a preventive effect on osteoporosis. The production of glutamate is closely related to the fermentation of lactic acid bacteria [134]. RANK/RANKL signaling and NFATc1 play crucial roles in osteoclast development. Hwangryun-haedok-tang (HRT), a blend of Chinese herbs including *Coptis japonica Makino*, *Phellodendron chinense Schneider*, *Gardenia jasminoides fructus*, and *Scutellaria baicalensis*, has been shown to inhibit NFATc1 expression and exert inhibitory effects on RANKL-induced osteoclast development. The fermentation of lactic acid bacteria enhances the inhibitory effects of HRT on postmenopausal osteoporosis [135].

## 6. Summary and Outlook

In recent years, consumers have paid more and more attention to the natural and functional nature of products. Currently, various products available on the market are heavily laden with additives and lack functional properties. The lactic acid bacteria fermentation of plant-based products gives rise to active ingredients that not only act as additives, such as preservatives and flavor enhancers, but also bestow upon these products enhanced

functionality. In addition, lactic acid bacteria fermentation can improve the utilization of plant-based by-products. Lactic acid bacteria fermentation plays an important role in the development and application of plant-based products.

However, the key technology of lactic acid bacteria fermentation of plant-based products still faces many challenges. Most of the current fermentations are at the laboratory research stage, resulting in a limited variety of flavors and weak probiotic functionality. In the future, researchers should work hard to continuously select excellent fermentation strains and improve fermentation methods, so as to realize the diversification of the flavor, nutrition, and function of fermented plant-based products. Another issue is that the current plant-based fermented products on the market are not significantly different in flavors from traditional beverages and lack purity. The wide price disparity between the two has resulted in limited consumer acceptance. Therefore, researchers should continuously improve the production process of plant-based fermentation products and improve the utilization rate of plant by-products, thereby reducing production costs. The interaction mechanism between the active substances and functional properties of fermented plant-based products is not yet well defined. Researchers can maximize the functionality of the products by studying the mechanisms of action of active substances in fermented plant-based products. It is doubtful whether the functional products can play the greatest probiotic role in the human body. Therefore, the fermented plant-based products need to continue to conduct in-depth animal experiments and clinical studies. Lastly, the application of lactic acid bacteria fixed on plant-based materials for fermentation is relatively limited due to insufficient technological maturity. Future research should focus on innovative applications and explore how the immobilized fermentation of lactic acid bacteria can maximize the production of fermented plant-based products.

Plant-based products and lactic acid bacteria are two important areas of development in the future of the food industry. The combined and mutually reinforcing development of these two parts is crucial for diversifying food options in terms of nutritional value, sensory qualities, and functional properties. This integrated approach represents the primary trend in future food development.

**Author Contributions:** Conceptualization, J.H.; methodology, C.C.; investigation, H.M.; visualization, L.W.; resources, X.F.; supervision, J.W.; writing—original draft preparation, X.Y.; writing—review and editing, Z.L. All authors have read and agreed to the published version of the manuscript.

**Funding:** This work was financially supported by the Key Area Research and Development Program of Guangdong Province (2020B020226008 and 2018B020206001).

**Institutional Review Board Statement:** Not applicable.

**Informed Consent Statement:** Not applicable.

**Data Availability Statement:** Not applicable.

**Acknowledgments:** This work was supported by South China Agricultural University.

**Conflicts of Interest:** The authors declare no conflicts of interest. The funders had no role in the design of the study; in the collection, analyses, or interpretation of data; in the writing of the manuscript; or in the decision to publish the results.

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
