# Peer review of "Effect of Lactic Acid Bacteria Fermentation on Plant-Based Products"

_fermentation, doi:10.3390/fermentation10010048_

Round 1

Reviewer 1 Report

Comments and Suggestions for Authors

In my opinion, the manuscript entitled: Effect of Lactic Acid Bacteria Fermentation on Plant-Based Foods is a complex and very useful review. Authors have discussed the main approaches, functionality, and application of lactic acid bacteria in the field of plant-based foods. 

The introduction provides enough information regarding the current state of the art. Aspects such as the effect of lactic acid bacteria (LAB) during fermentation on the main bioactive compounds (proteins, amino acids, vitamins, organic acids, phenolic compounds), the production of ESP, aroma volatile compounds and functional properties of LAB were very good highlighted. Flavor improvements, antioxidant activity effect, antimicrobial properties, anti-inflammatory function, hypoglycemic function, improvement of the immune function, positive effect on the intestinal flora regulation, anti-tumor function, anti-obesity effects, and osteoporosis were the main functional LAB properties.

I just have some small suggestion:

1.     Figure 1. The effect of lactic acid bacteria fermentation on plant-based foods, please explain the abbreviation used, under the figure. For instance, EPS, T-cells, PD-P1. In the figure you might use only the abbreviation, in order to avoid so much text in the figure. 

2.     Lines 18-19 please try not to repeat so many the words offers, offering. You may use instead of offers emphasize or highlight.

Thank you!

Comments on the Quality of English Language

 Minor editing of English language required

Author Response

Response to Reviewer 1 Comments

Reviewer 2 Report

Comments and Suggestions for Authors

Manuscript 2772114

Journal Fermentation

Title Effect of Lactic Acid Bacteria Fermentation on Plant-Based Foods

The review entitled “Effect of Lactic Acid Bacteria Fermentation on Plant-Based Foods” summarizes recent findings on the application of lactic acid bacteria for the fermentation of plant-based foods, and the changes related to nutritional, sensory and health-promoting features. The topic is not novel and several reviews are available in literature on this topic. The manuscript needs substantial revision. Please follow the comments in the file.

Comments on the Quality of English Language

Moderate changes are necessary

Author Response

Thank you very much for your comments and professional advice. These opinions help to improve the academic rigor of our article. We have carefully considered each comment and suggestion, and we are grateful for the opportunity to address them. Please find below our detailed responses to the reviewers' major revision points:

Major comment:

Which is the novelty of the review?

The author's answer:

We have revised the abstract and added some innovative points about this article. The details follow:

“In controlling obesity, cardiovascular disease, cancer, and other diseases, adjusting the diet may have a greater effect than taking drugs. Lactic acid bacteria have powerful metabolic characteristics and enzymes, which can improve the content of polyphenols, vitamins, amino acids and other active substances in fermented food. At the same time, it can enhance the antioxidant, anti-inflammatory, anti-obesity, and other functions of fermented plant-based foods. A combination of nutrition, health, and taste can be accomplished through the fermentation of plant-based foods by lactic acid bacteria. Dietary structure may be optimized and nutritional health levels could be enhanced with the help of fermented plant-based foods. However, the variety of plant-based foods used for lactic acid bacterial fermentation is currently limited, and research on post-fermentative impacts on health and disease prevention are not in-depth. This paper aims to introduce the process of fermenting plant-based foods with lactic acid bacteria to offer additional research methods for the efficient utilization of plant-based foods and plant-based waste. Simultaneously, this review examines the impact of lactic acid bacteria fermentation on the active substances and functional properties of plant-based foods. The goal is to direct researchers' attention to the functional properties of fermented plant-based food and encourage more extensive clinical and experimental research. Doing so can advance the shift towards healthier modern diet habits and the development of food therapy.”

Other comments: Please refer to the document for detailed reply.

Round 2

Reviewer 2 Report

Comments and Suggestions for Authors

Manuscript 2772114

Journal Fermentation

Title Effect of Lactic Acid Bacteria Fermentation on Plant-Based Foods

The review entitled “Effect of Lactic Acid Bacteria Fermentation on Plant-Based Foods” summarizes recent findings on the application of lactic acid bacteria for the fermentation of plant-based foods, and the changes related to nutritional, sensory and health-promoting features. The topic is not novel and several reviews are available in literature on this topic. The manuscript has been revised but still needs substantial revision. Please follow the comments in the file. 

Comments on the Quality of English Language

Extensive English revision is necessary.

Round 3

Reviewer 2 Report

Comments and Suggestions for Authors

Authors addressed large part of the reviewer's comments. English language should be revised throughout the manuscript (e.g., see the sentence at L168-173).

Comments on the Quality of English Language

English language should be revised throughout the manuscript